# Chondrogenic Differentiation of Adipose-Derived Stromal Cells Induced by Decellularized Cartilage Matrix/Silk Fibroin Secondary Crosslinking Hydrogel Scaffolds with a Three-Dimensional Microstructure

**DOI:** 10.3390/polym15081868

**Published:** 2023-04-13

**Authors:** Jing Zhou, Nier Wu, Jinshi Zeng, Ziyu Liang, Zuoliang Qi, Haiyue Jiang, Haifeng Chen, Xia Liu

**Affiliations:** 1Plastic Surgery Hospital, Chinese Academy of Medical Sciences and Peking Union Medical College, Beijing 100144, China; zhoujing@psh.pumc.edu.cn (J.Z.); zengjs@163.com (J.Z.); qizuoliang@psh.pumc.edu.cn (Z.Q.); jianghaiyue@psh.pumc.edu.cn (H.J.); 2State Key Laboratory of Pathogen and Biosecurity, Beijing Institute of Microbiology and Epidemiology, Beijing 100071, China; wnezymd@126.com; 3Department of Biomedical Engineering, College of Future Technology, Peking University, Beijing 100871, China; 1701111669@pku.edu.cn; 4Key Laboratory of Reconstruction for Superfacial Tissues and Organs, Beijing 100144, China

**Keywords:** cartilage tissue engineering, secondary crosslinking, decellularized cartilage extracellular matrix, silk fibroin, multi-channeled

## Abstract

Finding an ideal scaffold is always an important issue in the field of cartilage tissue engineering. Both decellularized extracellular matrix and silk fibroin have been used as natural biomaterials for tissue regeneration. In this study, a secondary crosslinking method of γ irradiation and ethanol induction was used to prepare decellularized cartilage extracellular matrix and silk fibroin (dECM-SF) hydrogels with biological activity. Furthermore, the dECM-SF hydrogels were cast in custom-designed molds to produce a three-dimensional multi-channeled structure to improve internal connectivity. The adipose-derived stromal cells (ADSC) were seeded on the scaffolds, cultured in vitro for 2 weeks, and implanted in vivo for another 4 and 12 weeks. The double crosslinked dECM-SF hydrogels exhibited an excellent pore structure after lyophilization. The multi-channeled hydrogel scaffold presents higher water absorption ability, surface wettability, and no cytotoxicity. The addition of dECM and a channeled structure could promote chondrogenic differentiation of ADSC and engineered cartilage formation, confirmed by H&E, safranin O staining, type II collagen immunostaining, and qPCR assay. In conclusion, the hydrogel scaffold fabricated by the secondary crosslinking method has good plasticity and can be used as a scaffold for cartilage tissue engineering. The multi-channeled dECM-SF hydrogel scaffolds possess a chondrogenic induction activity that promotes engineered cartilage regeneration of ADSC in vivo.

## 1. Introduction

Congenital disorders, trauma, or cancer can cause the malformation of cartilage tissue, leading to a psychological and financial burden on patients and their families. The repair of cartilage defects is a challenge for surgeons because of its poor intrinsic healing capacity and hypocellular and hypovascular nature [1,2]. Although multiple materials have been used to repair cartilage defects, including prosthesis [3], the implantation of nonabsorbable artificial materials [4,5], and autologous cartilage and chondrocyte implantation [6], they are either overly invasive or barely receive satisfactory results. The rapid development of cartilage tissue engineering offers a promising alternative for cartilage repair and regeneration [7,8,9,10].

Currently, decellularized cartilage extracellular matrix (dECM) is considered a promising biomaterial since it is a direct way to provide the complex composition of native tissue, which is difficult to reproduce using common biomaterials. Our previous study has demonstrated that dECM retained most ECM-related components while cellular components were removed [11]. The components of dECM from cartilage reveal chondro-inductivity for mesenchymal stem cells (MSC) and the potential for supporting new matrix synthesis [12]. In addition, dECM is generally conserved among species and is tolerated well even by xenogeneic recipients, so it has been widely studied and used in a variety of tissue regeneration applications [13,14]. However, it is difficult to directly manufacture dECM into a scaffold because of its weak mechanical properties and poor processability. Silk fibroin (SF), as a naturally occurring polymer, has emerged as an attractive option for tissue engineering applications, due to its good mechanical strength, easy processability, and good cell compatibility [15,16]. Incorporating dECM with SF presents a promising strategy to create hydrogel scaffolds for engineered cartilage formation [16].

However, there are still some deficiencies in the preparation of traditional dECM-SF hydrogel scaffold. As an advanced material form, polymer hydrogel is particularly similar to the extracellular matrix which is composed of biological macromolecules and can simulate the characteristics of biological tissue very well [8]. While mature engineered cartilage tissue can form on the surface of the hydrogel, it is difficult to find in the inner part of it, which might be due to the poor porosity that hinders nutrient exchange. The preparation of hydrogel scaffolds with multi-channeled structures requires the stiffness of the hydrogel material itself. In addition, the crosslinking method of the hydrogel is crucial to the mechanical properties of the biomaterials. At present, the traditional chemical, physical, and energy excitation crosslinking methods used to prepare hydrogels have their limitations, thus it is difficult to effectively regulate the mechanical properties, pore connectivity, and degradation time [17,18,19,20].

In this study, we introduced ethanol after γ-ray radiation and further established the “secondary network” by developing a versatile scaffold with desirable stiffness and biochemical properties [20]. On this basis, we fabricated a new multi-channeled hydrogel scaffold by the reverse molding technique to improve the interconnectivity and nutrient exchange. With this technique, we prepared a hydrogel scaffold material with low cytotoxicity and high stiffness and plasticity. We hypothesized that such a scaffold could promote the chondrogenic differentiation of adipose-derived stromal cells (ADSC) and facilitate engineered cartilage formation. This study aimed to test the biological cellular responses of new hydrogel scaffolds. ADSC were harvested and seeded onto scaffolds and then implanted into nude mice to compare their ability to form engineered cartilage in vivo.

## 2. Materials and Methods

### 2.1. Experimental Animals

Bama miniature pigs (8 weeks old, *n* = 4) and female BALB/C nude mice (6 weeks old, *n* = 40) were both purchased from Beijing Vital Step Experimental Animal Raising Farm (Beijing, China). All the experimental procedures and protocols for animal care were approved by the Animal Care and Experimental Ethics Committee of Peking Union Medical College. All the animal subjects received care following the “Guide for Care of Laboratory Animals”, as detailed by the National Institutes of Health.

### 2.2. Cell Culture

Adipose tissues were collected from the colli posterior fat regions of the pigs for ADSC isolation. As previously described [16], the ADSC were isolated and cultured in low-glucose (1000 mg/L) Dulbecco’s modified Eagle’s medium (L-DMEM, GE Healthcare Life Sciences, Piscataway, NJ, USA) supplemented with 10% fetal bovine serum (FBS, Thermo Fisher Scientific Life Sciences, Waltham, MA, USA), 100 U/mL penicillin, and 0.1 mg/mL streptomycin at 37 °C with 95% humidity and 5% CO_2_. Passage 3 ADSC were harvested for engineered cartilage regeneration.

### 2.3. Preparation of Decellularized Cartilage Extracellular Matrix and Silk Solution

DECM was prepared as previously described [21]. Cartilage pieces were harvested from pig ears and then washed and shattered in phosphate-buffered saline (PBS) containing 3.5% (*w*/*v*) phenylmethylsulfonyl fluoride (PMSF, Merck, Darmstadt, Germany) and 0.1% (*w*/*v*) EDTA (Sigma, Poole, UK) to inhibit protease activity. The subsequent treatments, unless otherwise stated, all involved protease inhibition. To obtain a cartilage slurry, small pieces of cartilage were blended in PBS and pulverized using a homogenizer (IKAT10, IKA). The homogenized cartilage tissue was blended in PBS and centrifuged in an eppendorf centrifuge 5810R (eppendorf, Hamburg, Germany) for 5 min at 2000 rpm. Then, the deposit was removed, and a new suspension was recentrifuged for another 5 min at 7000 rpm. After pulverization and differential centrifugation, cartilage microfilaments with diameters of approximately 500 nm to 5 mm were then incubated in 1% TritonX-100 in hypotonic Tris-HCI with gentle agitation for 12 h at 4 °C. After being thoroughly rinsed in PBS, the samples were digested with 50 U/mL deoxyribonuclease and 1 U/mL ribonuclease (Sigma-Aldrich) in 10 mM Tris-HCI (pH 7.5), with agitation for 12 h at 37 °C without protease inhibition. The decellularized cartilage matrix microfilaments were then washed intensively with PBS and made into a 6% (*w*/*v*) suspension.

The cocoons were boiled in a degumming solution (0.02 M Na_2_CO_3_) for 30 min and then washed thoroughly in double-distilled water to remove the sericin protein. After drying, the degummed silk was dissolved in a 9.3 M lithium bromide (LiBr) solution at 60 °C for 4 h. Then, the solution was dialyzed in a 3500 kD dialysis bag against double-distilled water. After removing the LiBr, the solution was centrifuged at 9000 rpm for 20 min at 4 °C to remove impurities. Then, the supernatant was freeze-dried overnight and made up to a 6.9% (*w*/*v*) silk solution [22].

### 2.4. Fabrication of Secondary Crosslinked dECM-SF Scaffolds

#### 2.4.1. Solid Hydrogel Scaffold Manufacturing

The preparation process of the scaffolds is demonstrated in Figure 1. The mixture consisted of 6.9% (*w*/*v*) SF solution and 6% (*w*/*v*) dECM suspension at volume ratios of 1:1 and was poured into a mold and radiated by γ-ray radiation using a 60Co radiation facility according to the 60 kGy total doses at 196 Gy/min at room temperature to form a primary hydrogel. Then, the hydrogel was soaked in anhydrous ethanol for 30 min to form a secondary hydrogel. The hydrogel was then washed in double-distilled water for 30 min 5 times to displace the ethanol. In the end, the hydrogel was freeze-dried overnight to obtain the dECM-SF hydrogel scaffolds. Pure silk solution was applied to obtain SF hydrogel scaffolds via the same procedure. 

The crosslinking of SF and dECM induced by γ-ray undergoes a mechanism of radical polymerization, which is similar to the γ-ray-induced crosslinking of collagen [23]. In detail, active groups such as hydroxyl radicals (·OH), proton radicals (·H), hydrated electrons (eaq-), and superoxide (O_2_^−^) were generated by irradiation in aqueous solution (Figure 2A (1)) and, thus, provoked a chain reaction in which the removal of H occurred on various amino acid residues on the polypeptide chains. The unpaired electron of the peptide radicals should stay on the α-C of the amino acid residues in order to form the most stable state (2). The combination of the polypeptide radicals resulted in chemical conjugation, which finally formed a crosslinking network between the SF and collagen from the ECM (3).

The treatment of ethanol induces a conformation change of the SF and dECM polypeptide chain. Basically, neighboring repeating sequences of GAGAGS align together to form hydrogen bonds. As a result, the secondary structure of the α-coil and random coil change into β-sheets (Figure 2B), which corresponds to better mechanical properties of the hydrogel.

#### 2.4.2. Fabrication of the Multi-Channeled Hydrogel Scaffold Using the Reverse Molding Process

PCL reverse molds were fabricated by a 3D-bioplotter (EnvisionTEC, Gladbeck, Germany). The PCL molds were designed by computer-aided design (CAD) in Unigraphics NX 11.0 software (Siemens PLM Software, USA). PCL powder (molecular weight 45,000; Sigma, St. Louis, MO, USA) of a pharmacological grade was used as the printing material. The extrusion pressure was 8.0 bar, and the printing speed was 1.2 mm/s. The temperature of the printing head and the bottom plate were 80 °C and 28 °C, respectively. The printed cylinders were 0.5 mm in diameter and 13.0 mm in length, with 0.8 mm spacing and a 0°/90° laydown pattern with successive stacking for 8 layers, resulting in the fabrication of a cuboid structure of 13.5 mm in length and 4 mm in height.

The solution consisting of dECM and SF was injected into the sacrificial mold and placed in a vacuum box for 12 h to remove bubbles within the mold and allow the fill of suspension. Then, the compound was radiated by γ-ray radiation and later soaked in anhydrous ethanol for 30 min to form a secondary hydrogel. Then, the hydrogel was immersed in dichloromethane for 12 h to remove the PCL components, leaving a multi-channeled scaffold. The scaffold was washed with distilled water for 30 min, and this was repeated three times to remove residual chemicals. Cuboid scaffolds with a length of 6.5 mm and a height of 4 mm were cut and stored at 4 °C. For the control group, multi-channeled SF scaffolds without dECM were fabricated using SF solution alone and the same method.

### 2.5. Characterization of the Scaffolds

#### 2.5.1. Fourier Transform Infrared Spectroscopy

The SF solution, dECM solution, SF hydrogel, and dECM-SF hydrogel were treated with liquid nitrogen to prevent protein denaturation and then lyophilized for 48 h. The samples were prepared by the KBr pellet method, and the mid-infrared spectra of the samples (wavenumber range 4000 cm^−1^–400 cm^−1^) were measured by a Fourier transform infrared spectrometer (FTIR, Nicolet is50, ThermoFisher) with a resolution of 2 cm^−1^. The results were analyzed using GraphPad Prism 8.

#### 2.5.2. Examinations of Water Absorption Capacity and Surface Wettability

By the immersion of dried scaffolds of known weight (Wd) in PBS at room temperature for 24 h, the swollen weight (Ws) of the scaffolds was measured after the removal of the redundant water on the surface. The water uptake of the scaffolds and the swelling ratio were determined according to the below formula [24].
Water uptake = [(Ws − Wd)/Ws] × 100%
Swelling ratio = (Ws − Wd)/Wd.

The surface wettability of the scaffolds was determined by the water contact angle for each specimen at room temperature. Briefly, the specimens were placed on the top of a stainless-steel base, a drop of MilliQ DMEM (10 μL) was added to the surface of the specimens, and an image was taken with a camera after 20 s had elapsed. The resulting images were analyzed using ImageJ (National Institutes of Health, Bethesda, MD, USA) to determine the water contact angle.

#### 2.5.3. Cytotoxicity Assay

To determine the cytotoxicity of the scaffolds, CCK-8 (Sigma) was used to assess cell proliferation. The leaching fluid of the hydrogel was obtained as previously described [25]. Briefly, ADSC (3 × 10^3^ in 200 μL suspension) were seeded into 96-well plates. After incubation with leaching medium and control medium (DMEM (Gibco, Billings, MT, USA) containing 10% FBS (Gibco)) for 1, 3, 5, and 7 days, the cell samples were rinsed in Hanks salt solution (Gibco). Then, 100 μL DMEM medium containing 10% FBS and 10% CCK-8 solution was added to each well. After 1 h of additional incubation at 37 °C, the reaction solution was measured by a microplate reader (Wallac Victor 1420, Perkin Elmer Life Sciences, Waltham, MA, USA) at a wavelength of 450 nm. The optical density (OD) value was proportional to the number of cells. Five replicates were considered per group. The results were obtained in triplicate from three separate experiments.

### 2.6. Preparation of Cell-Scaffold Constructs

The hydrogel scaffolds were pre-incubated in DMEM supplemented with 10% FBS for 12 h in an incubator. ADSC at Passage 3 were collected and suspended in DMEM culture medium at a density of 1 × 10^7^ cells/mL, followed by seeding onto the scaffold (0.1 mL per construct). The cell-scaffold constructs were kept in an incubator for up to 4 h to allow for complete adhesion of the cells to the scaffolds, and then a culture medium without chondrogenic induction factors was added to cover the construct. Medium changes were performed every 2–3 days as needed. After in vitro culture for 2 weeks, the cell-scaffold constructs were implanted in vivo in nude mice.

### 2.7. Immunofluorescent Staining

To detect the remains of the cells, samples of dECM-SF and multi-channeled dECM-SF (MC dECM-SF) hydrogel were examined under fluorescence microscopy after DAPI (4′,6-diamidino-2-phenylindole) staining. For cell viability, the scaffolds were incubated with live/dead double staining solution (Sigma, USA) for 30 min and then observed by confocal microscopy (Leica, Heidelberg, Germany).

### 2.8. Scanning Electron Microscopy (SEM) Examination 

To observe the ultrastructure of the hydrogel scaffolds, the samples were coated with gold-palladium and examined under a scanning electron microscope (SEM; Hitachi S-520, Hitachi, Japan). The interior microstructures of the scaffolds were investigated using ImageJ (US National Institutes of Health, Bethesda, MD, USA). The distribution of the pore sizes, as well as the porosity, was assessed. The channels formed by PCL reverse mold were not included in the pore sizes and porosity. To examine the cell adhesion, the cell-scaffold constructs were prepared for SEM examination after 7 days of in vitro culture. The sample was fixed with 2.5% glutaraldehyde at 4 °C for 12 h then dehydrated using a graded ethanol series. A cross-section of the dried sample was coated with gold-palladium and examined under SEM.

### 2.9. Histological and Immunohistochemical Staining

The samples were fixed in 4% paraformaldehyde, embedded in paraffin, and then sectioned into 5 µm sections. The sections were stained with hematoxylin and eosin (H&E) to evaluate their structures and stained with Safranin O-Fast Green to visualize glycosaminoglycan (GAG) deposits. The expression of type II collagen was detected by mouse anti-human type II collagen monoclonal antibody (1:200; Abeam, Cambridge, MA, USA), followed by horseradish peroxidase (HRP)-conjugated anti-mouse secondary antibody (1:50, Dako, Denmark) and color development with diaminobenzidine tetrahydrochloride (DAB, Dako, Denmark). To detect the existence of DNA, samples of the dECM-SF scaffold and MC dECM-SF scaffold were examined under fluorescence microscopy after DAPI (4′,6-diamidino-2-phenylindole) staining. For cell viability, the scaffolds were incubated with live/dead double staining solution (Sigma, USA) for 30 min and then observed by confocal microscopy (Leica, Heidelberg, Germany).

### 2.10. Quantitative PCR Analysis 

The expression of collagen type II (Col 2), SOX-9, and aggrecan (Acan) was determined through quantitative PCR (qPCR). The total RNA was extracted using Trizol Reagent (Invitrogen, Camarillo, CA, USA), and complementary DNA was synthesized from 2 μg total RNA per sample using a First cDNA synthesis kit (Fermentas Life Sciences, York, UK). To perform quantitative real-time PCR, the target gene expression was analyzed using a Power SYBR Green PCR master mix (Applied Biosystems, Foster City, CA, USA) on a real-time thermal cycler (Stratagene Mx3000PTM QPCR System, La Jolla, CA, USA). qPCR was conducted in triplicate for each sample and experiments were repeated on three tissue samples. The expression level of each targeted gene was calculated by the 2^−ΔΔCt^ method and normalized to that of GAPDH. The primer sequences are listed in Table 1.

### 2.11. Statistical Analysis

All the quantitative data are presented as the mean ± standard deviation (SD). The t-test was used to analyze the difference between different groups. A value of *p <* 0.05 was considered statistically significant.

## 3. Results

### 3.1. Properties of the Hydrogel Scaffolds

The dECM-SF and SF hydrogel scaffold was solid disk-shaped, and the surface was relatively smooth. The multi-channeled scaffold presented as cube-shaped and the surface was relatively rough and grid-like. The microstructure of the hydrogels was characterized by scanning electron microscopy (SEM). The results showed that there was a homogenous microporous structure with less than 200 um in diameter in the SF scaffold, while the micropores were flat in the dECM-SF scaffold. In the multi-channeled group, there was a larger channeled structure formed in addition to micropores (Figure 3A). The dECM-SF scaffold had a similar pore size distribution as the SF scaffold (117.63 ± 15. 99 μm vs. 105.59 ± 20.9 µm, *p* > 0.05). The mean diameter of the pores in the MC dECM-SF and multi-channeled SF (MC SF) scaffolds was 114.88 ± 12.05 µm and 106.58 ± 11.24 μm, respectively, with no significant difference observed (*p* > 0.05). In addition, both the solid dECM-SF group and MC dECM-SF group showed similar porosity (61.11 ± 3. 99% vs. 64.07 ± 4. 7%, *p* > 0.05) (Figure 3B,C). These results demonstrated that the multi-channeled structure improved the water absorption capacity of the solid hydrogel scaffold. The surface wettability of the multi-channeled scaffolds and solid scaffolds was analyzed using a contact analyzer. As shown in Figure 3D, the DMEM added to the surface of the scaffold was absorbed in the MC group after 20 s, while some liquid was left on the surface of the solid scaffold with a contact angle of 117 ± 3.9°. The multi-channeled hydrogel showed a better swelling ratio than the solid hydrogel (21.09 ± 6.67 vs. 9.55 ± 3.53, *p* < 0.05) as well as water uptake (95.05 ± 1.68% vs. 89.62 ± 3.29%, *p* < 0.05) (Figure 3E,F). Cell residues in the dECM were detected by immunofluorescence. After DAPI staining, no positive nuclear blue fluorescence was detected within the scaffolds containing dECM, suggesting that no cells remained after the decellularizing treatment (Figure 3G). 

The FTIR results showed the conformational positions and changes of the SF and dECM before and after crosslinking. As shown in Figure 3H, the positions of the main peaks of the SF hydrogel samples appeared on 1626 cm^−1^ (amide I band), 1518 cm^−1^ (amide II band), 1238 cm^−1^ (amide III band), and 3280 cm^−1^ (N–H stretching), which had no significant differences with the positions of main peaks of the SF solution [26]. The positions of the main peaks of the dECM-SF hydrogel samples appeared on 1633 cm^−1^ (amide I band), 1545 cm^−1^ (amide II band), 1238 cm^−1^ (amide III band), and 3280 cm^−1^ (N–H stretching), which had no significant differences with the positions of the main peaks of the dECM suspension [27]. The transmittance spectra of the dECM-SF hydrogel showed slight variations of the main peaks, indicating that an interaction occurred during the crosslinking process between the dECM and SF.

Furthermore, the cytotoxicity of the hydrogel scaffold was investigated through ADSC culture in vitro. The ADSC cultured in the leaching medium of the scaffold displayed a similar proliferation rate and viability rate as the ADSC cultured in DMEM medium at all the time points (*p* > 0.05), indicating negligible cytotoxicity of both the solid and multi-channeled hydrogel scaffolds (Figure 3I and Appendix A). 

### 3.2. Cytocompatibility In Vitro 

After 7 days of culture in vitro, SEM examination revealed good cell attachment on the scaffolds in all the groups, and there were extracellular matrices produced, indicating good cytocompatibility of the hydrogel scaffolds. The live/dead double staining results showed that most seeded cells survived (green fluorescence) and there were more cells in the MC dECM-SF scaffold (Figure 4).

### 3.3. Gross View and Histology Evaluation of Engineered Cartilage

To evaluate whether mature cartilage could be achieved in vivo, after 2 weeks of culture in vitro, cell-scaffold constructs were implanted into nude mice for another 4 and 12 weeks. As shown in Figure 5, at 4 weeks post-implantation, a relatively dark-colored neo-tissue with a relatively smooth surface was formed in each group. At 12 weeks, the engineered cartilage in the dECM-SF and MC dECM-SF groups became glossy and white-colored compared to those in the other groups (Figure 6).

Histological analysis showed that after 4 weeks of implantation in vivo, cartilage-like tissues containing lacuna structures were observed in both the dECM-SF scaffold and MC dECM-SF scaffold groups. The expression of cartilage extracellular matrix was also confirmed by Safranin O-Fast Green staining and type II collagen immunohistological staining (Figure 5). By 12 weeks, H&E staining showed much more densely deposited cartilage matrices with apparent lacuna structures in the dECM-SF and MC dECM-SF scaffold groups. In the MC SF scaffold group, an obvious cartilage lacuna structure was also observed. Safranin O-Fast Green staining and type II collagen immunostaining indicated a higher level of collagen maturation in the MC dECM-SF scaffold group as opposed to the dECM-SF scaffold and MC SF scaffold groups (Figure 6). 

### 3.4. Gene Expression of Engineered Cartilage

To further clarify the roles of the three-dimensional multi-channeled structures and dECM components on the chondrogenic differentiation of ADSC, qPCR was performed to examine the gene expression related to cartilage. First, the role of three-dimensional multi-channeled structures on chondrogenic differentiation was examined. The results showed that at 4 weeks, the expression of Col 2 in the dECM-SF group was higher than that in the MC dECM-SF group, and there was no significant difference in ACAN expression, while the expression of Sox9 in the MC dECM-SF group was higher than that in the dECM-SF group (Figure 7). Until 12 weeks after implantation, the expression of Col 2, Sox-9 and ACAN in the MC dECM-SF group were all significantly higher than that in the solid dECM-SF scaffold (*p* < 0.01). These results indicated that dECM was important for ADSC chondrogenesis. While in the SF scaffolds, the expression of cartilage-related genes was higher in the multichannel scaffold than that in the solid scaffold from 4 weeks after implantation. This result demonstrated that the three-dimensional multi-channeled structure could promote chondrogenic differentiation of ADSC (Figure 7). 

Second, the induction of dECM in scaffolds on chondrogenic differentiation was detected. The gene expression of Col 2, Sox-9, and aggrecan was significantly higher in the scaffold containing the dECM component than in the hydrogel without the dECM component. This result demonstrated that the dECM component within the scaffold also could induce chondrogenic differentiation of ADSC (Figure 8).

## 4. Discussion

Hydrogel can be engineered to resemble the extracellular environment of different tissues in ways that enable their use in stem cell and cancer research, cell therapy, tissue engineering, and in vitro diagnostics. For regenerative medicine, hydrogel is an ideal type of scaffold material because hydrogel could be used to hierarchically organize cells into tissue-like structures [8,20,23]. Additionally, hydrogel could mediate multicellular morphogenesis by spatial and temporal presentations of architectural and/or molecular cues engineered in them [8]. In this study, dECM and SF were selected in combination to make the biomimetic hydrogel scaffolds for cartilage regeneration. Cartilage ECM is maintained by chondrocytes, within which there are growth factors and cytokines that regulate cell migration, colonization, and differentiation. For engineered cartilage, the natural cartilage ECM niche has inherent biochemical stimuli for chondrogenic differentiation of mesenchymal stem cells (MSC), which are difficult to replicate with synthetic biology [8]. SF is an attractive option for tissue engineering applications owing to its outstanding mechanical strength, easy processability, and good cytocompatibility. The combination of dECM and SF in manufacturing engineered cartilage scaffolds provides not only a cartilage-inducing environment for ADSC but also a sufficient support structure for tissue growth [28]. However, it is a challenge for crosslinking techniques to manufacture complex hydrogel systems for tissue engineering due to cytotoxicity and limited mechanical strength. 

At present, physical approaches, chemical treatment, and energy excitation are commonly adopted as crosslinking strategies [17,18,19]. Basic physical crosslinking methods are easy to operate without adding any crosslinking agents, but the random β-sheet crystal structures endow the products with poor mechanical strength and brittleness [17]. Although chemical crosslinking can overcome the brittleness of physically crosslinked products, bio-unfriendly components have to be introduced to improve their mechanical properties [18]. Energy excitation overcomes the limitation of chemical crosslinking, presenting as a highly efficient and bio-friendly procedure to manufacture scaffolds with a satisfactory structure [19]. Compared with ultraviolet and electron beam radiation, γ-ray radiation possesses a high-intensity penetrating ability which enables it to penetrate thick samples evenly and achieve a uniform inner structure. However, this method still has a problem, i.e., the stiffness of the hydrogel is relatively weak [29]. However, for cartilage engineering, the plasticity and stiffness of the hydrogel are crucial. To improve the mechanical properties and obtain a stiff hydrogel scaffold, a secondary crosslinking method was introduced in this study, as previously described [20]. The secondary crosslinking technique is one of the most commonly used methods to improve the stiffness of hydrogels by increasing the number of covalent bonds in the hydrogels [30,31]. Physical crosslinking via ethanol treatment, which constructs physical networks based on an γ-ray radiation crosslinking network, could transform those “soft” hydrogels produced by γ-ray radiation into tough hydrogels. The FTIR results showed that the positions of the main peaks of dECM had no significant change after secondary crosslinking, which demonstrated that the double crosslinking method did not destroy the dECM components in the hydrogel. In addition, the hydrogel scaffold had good cytocompatibility, and insignificant cytotoxicity was detected in the CCK-8 testing and live/dead staining assay. Therefore, these double crosslinked hydrogels can preserve functional proteins and show application prospects in tissue engineering.

The nutrient exchange capacity is another important factor of the hydrogel scaffold for cartilage engineering [29]. The solid hydrogels directly obtained by the secondary crosslinking method are relatively dense, and it is difficult for the ADSC to evenly distribute within the scaffold. Cartilage-like tissues were only observed on the surfaces in the solid hydrogel groups with loose structures in the central parts of the constructs. Based on the sufficient mechanical strength of the secondary crosslinked hydrogel, we adopted the reverse molding technique to create a three-dimensional multi-channeled structure within the solid hydrogel scaffold to improve internal connectivity. In this way, we manufactured “paths” to promote the nutrient exchange of the hydrogel. As shown in the results of the water absorption capacity and surface wettability assay, the multi-channeled structure significantly improves the water absorption capacity of the hydrogel, which could facilitate the cell seeding process. As expected, the in vivo results showed the neo-cartilage in the MC dECM-SF group was more homogenous than that in the solid dECM-SF group, with more mature lacuna structures and a denser extracellular matrix, suggesting that the multi-channeled structure provides a more conducive environment for the chondrogenesis of ADSC. 

Multipotent ADSC with abundant resources can differentiate into osteoblasts, chondrocytes, and adipocytes and have been used for cartilage engineering [32]. One major problem of cartilage engineering with ADSC is the low efficiency of chondrogenic differentiation. Studies have demonstrated that growth factors are required for directing chondrogenic differentiation [32]. A previous study showed that acellular cartilage sheet co-culture with mature chondrocytes could induce chondrogenic differentiation without supplemented growth factors because of the growth factors in the extracellular matrix, including TGF-b1, IGF-1, and BMP-2 [14]. Our results show that the hydrogel containing the dECM component has a pronounced chondrogenic induction effect with better cartilage-like tissue formation than the hydrogel without dECM. The ADSC-derived chondrocytes in the dECM scaffolds had a higher expression of COL2A1, SOX-9, and ACAN compared to those in the non-dECM scaffolds. Interestingly, neo-cartilage formation was also observed in the MC SF group, while there is almost no cartilage formation was observed in the solid SF group. It is known that silk fibroin has no chondrogenic induction effect [24]. This result indicated that a three-dimensional multi-channeled structure may exhibit chondrogenic induction of ADSC. However, the mechanism of the three-dimensional topography on the chondrogenic differentiation of ADSC is not fully understood [9]. The formation of cell spheroids has been reported to induce the differentiation of MSCs into chondrocyte-like cells [32,33]. We hypothesize that due to the three-dimensional structure of the multi-channeled hydrogel, the ADSC-differentiated chondrocytes aggregated into cell spheroids in the scaffolds, suggesting that this multi-channeled structure may enhance chondrogenesis. The pattern of mRNA expression further validates the role of the MC structure in chondrogenesis. The MC SF group had higher mRNA expression of COL2A1, SOX-9, and the ACAN than the solid SF group. This finding suggests that the dECM components and spatial cues had important influences on the gene expression and protein production of ADSC-differentiated chondrocytes.

There are some limitations in this study. The γ-ray radiation crosslinking technology combined with ethanol could regulate the mechanical properties of the hydrogels by inducing conformational transition. Based on the sufficient stiffness of the material, we fabricated the hydrogel with an internal multi-channeled structure for the first time and further confirmed that the ECM component and MC structure of the hydrogel could induce the chondrogenic differentiation of ADSC. Nevertheless, as a preliminary study, the effect of hydrogels with different mechanical strengths on chondrogenesis had not been studied. In addition, the scaffold architecture was strongly correlated with stem cell fate. We found that the MC hydrogel could enhance ADSC toward chondrogenesis. However, we have not yet performed an investigation into the mechanism. Our next research objectives are to optimize the crosslinking scheme and prepare the hydrogel scaffolds with the best mechanical properties. On the other hand, we aim to further define the mechanism of the microenvironment within the MC hydrogel scaffold that favors chondrogenic differentiation of MSCs.

## 5. Conclusions

In this study, we combined the secondary crosslinking method and reverse molding technique to construct a new MC dECM-SF hydrogel scaffold. These hydrogel scaffolds have good biocompatibility and water absorption capacity. The in vivo experiments demonstrated that the dECM component and multi-channeled structure of the hydrogel scaffold were favorable for the chondrogenic differentiation of ADSC. This study provides a sound rationale to further explore a potential mechanism of scaffold-induced chondrogenic differentiation of MSCs.

## Figures and Tables

**Figure 1 polymers-15-01868-f001:**
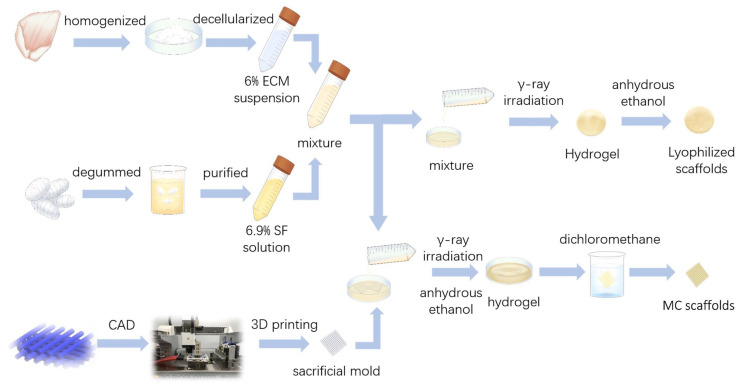
Graphical representation of the hydrogel scaffold preparation procedure.

**Figure 2 polymers-15-01868-f002:**
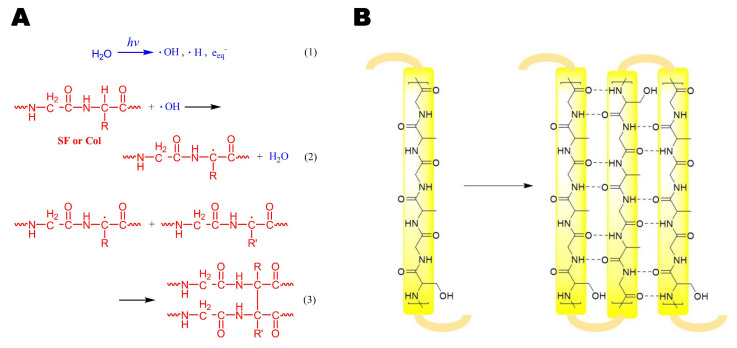
Mechanism of the γ-ray and ethanol crosslinking of SF-ECM hydrogel. (**A**). Mechanism of the γ-ray crosslinking. Active groups such as hydroxyl radicals were first generated from water by irradiation (1), after which polypeptide chains were attacked and turned to radicals (2). These peptide radicals could finally combine to form a crosslinking network (3). (**B**). Mechanism of ethanol crosslinking.

**Figure 3 polymers-15-01868-f003:**
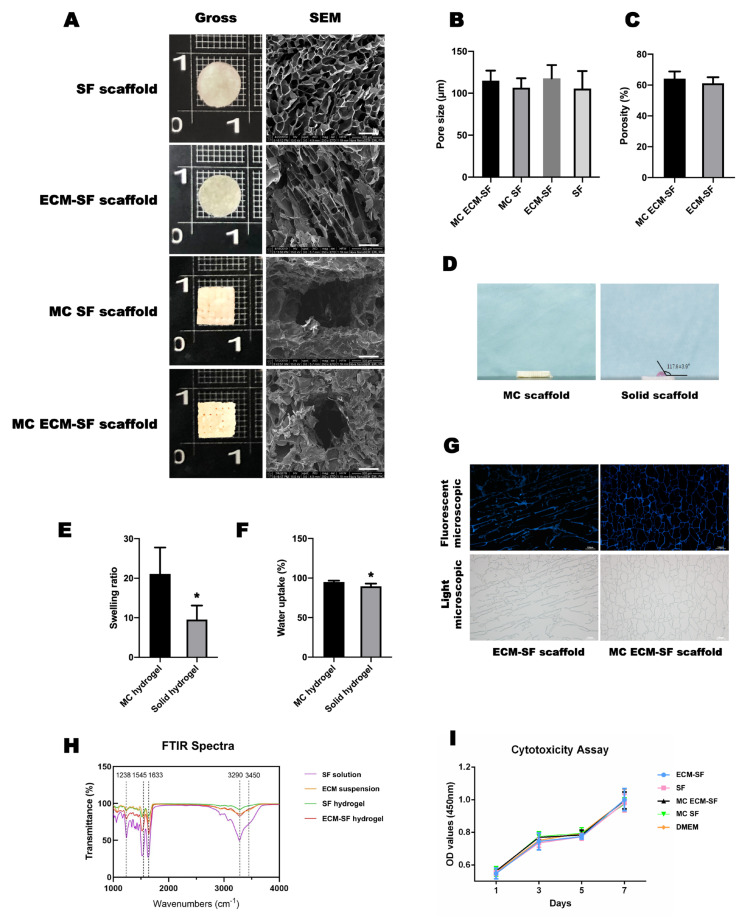
Structure and characterization of the scaffolds. (**A**). Gross view and ultrastructure of scaffolds, bar = 200 μm. (**B**). Pore size. (**C**). Porosity. (**D**). Surface wettability. (**E**). Swelling index of hydrogel scaffolds. (**F**). Water uptake. (**G**). DAPI and light microscopy images, bar = 100 μm. (**H**). FTIR spectra. (**I**). Cytotoxicity assay. “*” indicates a significant difference between the two groups with *p* < 0.05. Original magnification: ×250.

**Figure 4 polymers-15-01868-f004:**
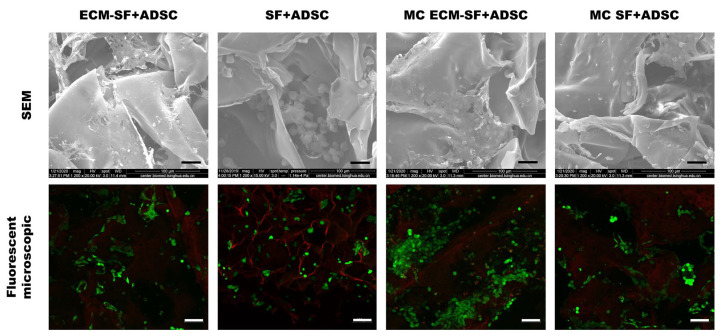
The cytocompatibility of the hydrogel scaffolds in vitro. SEM, bar = 30 μm. Fluorescence microscopy, bar = 100 μm.

**Figure 5 polymers-15-01868-f005:**
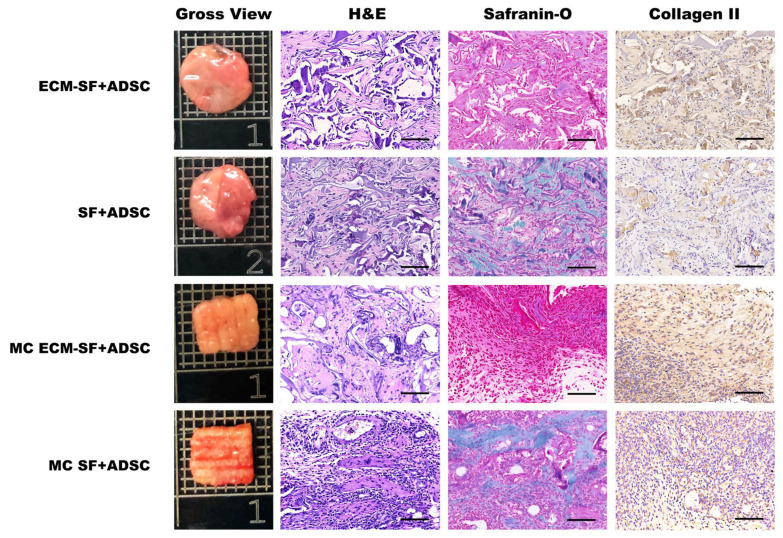
Gross view and histological examinations of engineered cartilage at 4 weeks in vivo. Bar = 20 μm.

**Figure 6 polymers-15-01868-f006:**
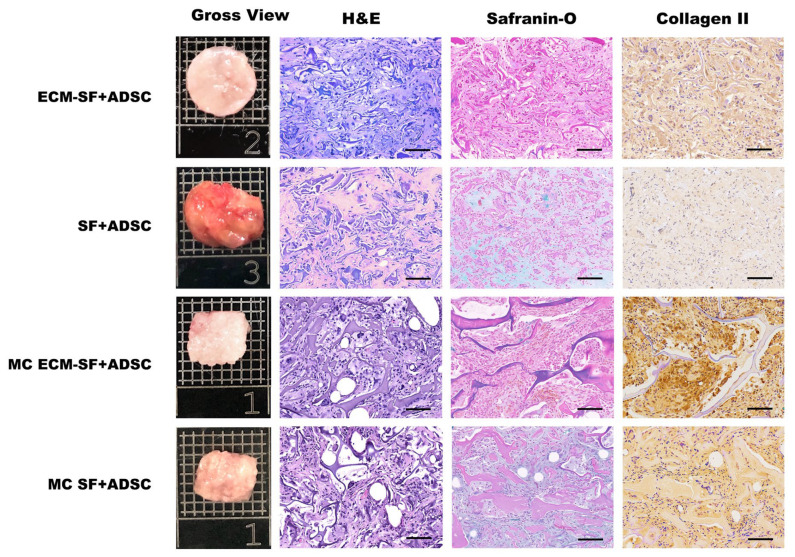
Gross view and histological examinations of the regenerated cartilage in nude mice for 12 weeks. Bar = 20 μm.

**Figure 7 polymers-15-01868-f007:**
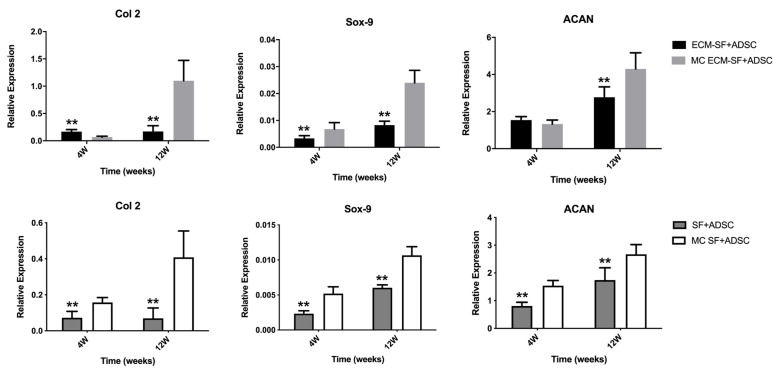
mRNA expression of chondrogenic-related genes examined with qPCR in solid hydrogel and MC hydrogel. “**” indicates significant difference between the two groups with *p* < 0.01.

**Figure 8 polymers-15-01868-f008:**
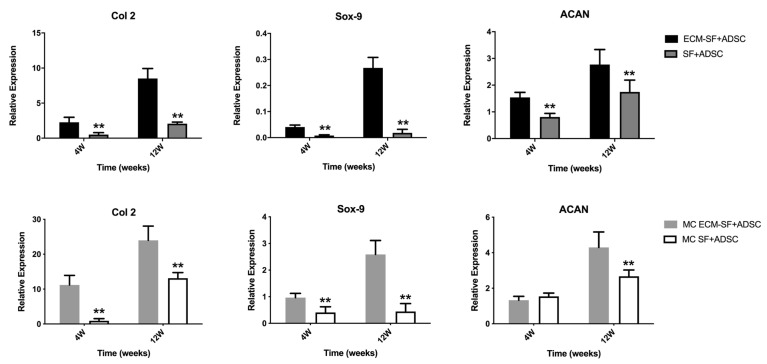
Comparison of the chondrogenic induction effects of hydrogels with or without dECM by qPCR. “**” indicates a significant difference between the two groups with *p* < 0.01.

**Table 1 polymers-15-01868-t001:** Reverse transcription polymerase chain reaction primer sequences.

Gene	Primer Sequences	
	Forward	Reverse
GAPDH	TCAAGCTCATTTCCTGGTACGA	CTCTTGCTGGGGTTGGTGGT
COL II	GAAGGATGGCTGCACGAAAC	GTCCACACCGAATTCCTGCT
Sox-9	AGAATCCTGGCATTTAAACCATA	TCACAGAGAAAACAAAAGGTTGG
Aggrecan	CGTGTAAAAAGGGCACAGTGG	GCACCAGGGAATTGATCTCGT

Abbreviation: GAPDH, glyceraldehyde 3-phosphate dehydrogenase.

## Data Availability

The datasets generated during and/or analyzed during the current. study are available from the corresponding author upon reasonable request.

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
