# Peer review of "Chondrogenic Differentiation of Adipose-Derived Stromal Cells Induced by Decellularized Cartilage Matrix/Silk Fibroin Secondary Crosslinking Hydrogel Scaffolds with a Three-Dimensional Microstructure"

_polymers, 2023, doi:10.3390/polym15081868_

Round 1

Reviewer 1 Report

This manuscript is interesting research in the field of the novel scaffolds for cartilage tissue engineering. The authors represent an effective strategy by combined secondary crosslinking method of γ irradiation and ethanol induction and reverse molding technique to obtain novel scaffold based on decellularized cartilage extracellular matrix (dECM) and silk fibroin (SF). The adipose-derived stomal cells were seeded on the scaffolds and implanted in vivo. The scaffolds were tested in terms of their chemical composition, swelling, surface wettability, cytotoxicity, morphology, biocompatibility and obtained results confirming favorable properties for their application as scaffolds. Among that, multi-channeled dECM-SF scaffolds possess a chondrogenic induction activity that facilitate in vivo engineered cartilage regeneration. This topic should be of interest for the community of medicine, pharmaceutical, and biomaterials science and the authors did good work. The manuscript is organized and written well with adequate illustrations. However, authors need to include following comments before publication.

1.      Limitation of the work, novelty and contributions should be highlighted more.

2.       More chemistry is required to understand the molecular interaction between dECM and SF. The only confirmation of the interaction between dECM and SF is transmittance spectra which showed variations of the main peaks indicating some interactions but not their identification.

3.      The authors mention the importance of mechanical strength of the scaffolds for cartilage engineering (L68-69, 369-372…) but for this study they did not examine mechanical properties of the scaffolds.

Author Response

Response to Reviewer 1 Comments

Comment 1: Limitation of the work, novelty and contributions should be highlighted more.

Response 1: Thank you very much for your valuable comments and suggestions. The limitations and novelty and contributions have been highlighted at the end of the Introduction and Discussion. The information is added in the Discussion as “There are some limitations to this study. … to further define the mechanism of the microenvironment within the MC hydrogel scaffold that favors chondrogenic differentiation of MSCs”.

The novelty is highlighted in the Introduction as “With this technique, …”.

Comment 2: More chemistry is required to understand the molecular interaction between dECM and SF. The only confirmation of the interaction between dECM and SF is transmittance spectra which showed variations of the main peaks indicating some interactions but not their identification.

Response 2: Thanks for the comment. The interation between SF and dECM induced by γ-ray undergoes a mechanism of radical polymerization, which is similar to the γ-ray induced crosslinking of collagen (see reference 23). The mechanism of the γ-ray and ethanol crosslinking of SF-ECM hydrogel has been illustrated in Figure 2. The information is added in the Materials and Methods as “ The crosslinking of SF and dECM induced by γ-ray undergoes a mechanism of radical polymerization, which is similar to the crosslinking of collagen induced by γ-ray [23]. … which finally formed a crosslinking network between SF and collagen from ECM (3).

Treatment of ethanol induces a conformation change of SF and dECM polypeptide chain. … which corresponds to better mechanism properties of the hydrogel”.

Comment 3: The authors mention the importance of mechanical strength of the scaffolds for cartilage engineering (L68-69, 369-372…) but for this study they did not examine mechanical properties of the scaffolds.

Response 3: Thank you for your advice. The mechanical strength is of great importance to fabricate scaffolds with special structure. In our study, we improved the stiffness of the ECM through secondary cross-linking with SF and successfully manufactured the multi-channel three-dimensional structure and verified the feasibility of this material for cartilage tissue engineering. “The secondary cross-linking technique is one of the most commonly used methods to improve the stiffness of hydrogels by increasing the number of covalent bonds in hydrogels [30,31]”.

Indeed, the mechanical properties of scaffolds are critical for cartilage tissue engineering. In this preliminary study, the mechanical properties were not explored in depth. Clarifying the mechanical properties of hydrogels and the effects of materials with different mechanical properties on chondrogenesis is also a goal of our next research.

The information is added in the Discussion as “There are some limitations to this study. … to further define the mechanism of the microenvironment within the MC hydrogel scaffold that favors chondrogenic differentiation of MSCs”.

Reviewer 2 Report

Journal: MDPI - Polymers

Manuscript

Title:  "Chondrogenic Differentiation of Adipose-Derived Stromal Cells Induced by Decellularized Cartilage Matrix/ Silk Fibroin Secondary Crosslinking Hydrogel Scaffolds with Three-Di-4 mensional Microstructure"

Author(s): Jing Zhou, Nier Wu, Jinshi Zeng, Ziyu Liang, Zuoliang Qi, Haiyue Jiang, Haifeng Chen, Xia Liu

Reviewer Comments to Author(s)

Recommendation: Major Revisions

This manuscript presents a comprehensive study of decellularized cartilage extracellular matrix and silk fibroin 16 (dECM-SF) hydrogels with biological activity. The manuscript corresponds to a well-structured hypothesis, and research work on biological cellular responses of new hydrogel is presented. The author(s) could think of the following simple corrections.

1.     There is a separate paragraph named 3.2 Figures, Tables and Schemes. It is extremely difficult in reading to gather all of these pieces of information at one place. The author’s need to provide the results (figures, tables or schemes) wherever they are first mentioned within the Part of Results for the reader to follow the research. Example, the scheme of Figure 1 should be provided when mentioned at paragraph 2.4

2.     Line 320 is probably from the guidelines to authors for submission but forgotten by the authors.

3.     Figure 2 I, Cytotoxicity assay results are presented in OD values but it is highly recommended to provide either cytotoxicity or viability in percentage rate compared to control cells. Which was the positive and negative control cells used in every cellular experiment?

4.     Lines 179-180, the authors say that they followed five replicates in triplicate separate experiments. Is this correct? This means that for each sample they performed 5 replicates in triplicate, equal to 15 evaluations for each sample? Please clarify the replicated tests and the triplicate meaning. Line 174 between the numbers of the days the space has been forgotten. Line 184 the 7 is superscript

5.     Paragraphs and section 2.1 up to 2.9 are plain but in part 3 paragraphs and sections are in Italic. The authors need to follow one style of presentation. Moreover, paragraph 3.1 needs to have a title not the example of the guidelines.

6.     In Figure 2 G the DAPI results of ECM-SF scaffold and the results of light microscopy seem to have different orientation. The lines of the light microscopy image do not follow the lines of the DAPI-blue fluorescence. In all Figures more detailed description need to be provided for the visualized results. For example, in line 258-259 the authors refer to Figure 2G results by mentioning scaffolds containing dECM, however this is not provided clearly either in the Figure or within the description. The authors state at line 215 that they use DAPI for evaluating DNA existence and at line 258 that there is no blue-fluorescence, however in Figure 2G there is blue fluorescence. Figures 2G, H, and I are too small to be clearly studied and evaluated.

7.     The authors should connect their results with the discussion and mention the specific figures when appropriate. For example, lines 376-377 the authors say that “As shown in the results, the double cross-linking method did not destroy the 376 dECM components in the hydrogel”. How is this supported by their results? More explanations need to be provided and proper connection of results and discussion.

Author Response

Response to Reviewer 2 Comments

Comment 1: There is a separate paragraph named 3.2 Figures, Tables and Schemes. It is extremely difficult in reading to gather all of these pieces of information at one place. The author’s need to provide the results (figures, tables or schemes) wherever they are first mentioned within the Part of Results for the reader to follow the research. Example, the scheme of Figure 1 should be provided when mentioned at paragraph 2.4.

Response 1: First, we sincerely thank the reviewer for the valuable and noteworthy comment to our study. And sorry for the inconvenience. The figures, tables and schemes are now provided wherever they are first mentioned within the part of Results.

Comment 2: Line 320 is probably from the guidelines to authors for submission but forgotten by the authors.

Response 2: Thank you very much for your valuable advice. L320 and the separate paragraph are inconvenience for readers to follow. The paragraph has been deleted. The figures, tables and schemes in section 3.2 have been cited in the main text accordingly.

Comment 3: Figure 2 I, Cytotoxicity assay results are presented in OD values but it is highly recommended to provide either cytotoxicity or viability in percentage rate compared to control cells. Which was the positive and negative control cells used in every cellular experiment?

Response 3: ADSCs cultured in control medium (DMEM (Gibco) containing 10% FBS (Gibco)) were severed as blank control, while the other four groups of cells cultured in leaching medium were the experiment groups. Thank you for your suggestion. We presented cytotoxicity assay results in OD values in order to show the growth of cells and the result showed insignificant differences among all groups. The viability in percentage rate compared to control cells on the seventh day will be provides as the supplementary material.

Supplement 1. ADSC cultured in the leaching medium of the scaffold displayed a similar viability rate as the ADSC cultured in DMEM medium on the seventh day (p > 0.05).

Comment 4: Lines 179-180, the authors say that they followed five replicates in triplicate separate experiments. Is this correct? This means that for each sample they performed 5 replicates in triplicate, equal to 15 evaluations for each sample? Please clarify the replicated tests and the triplicate meaning. Line 174 between the numbers of the days the space has been forgotten. Line 184 the 7 is superscript.

Response 4: Sorry for the confusion. We conducted three separate cytotoxicity experiments. In each experiment, there are 5 samples of ADSCs in each group.  The samples were tested on first, third, fifth and seventh days. And the same experiment was conducted three times. This means that for each group we performed 5 replicates in triplicate. Revision has been made “Five replicates were considered per group. The results were obtained in triplicate from three separate experiments”. Line 174, the spaces between the numbers of the days have been added. Line 184 the 7 is now superscript. The manuscript has been gone through and edited.

Comment 5: Paragraphs and section 2.1 up to 2.9 are plain but in part 3 paragraphs and sections are in Italic. The authors need to follow one style of presentation. Moreover, paragraph 3.1 needs to have a title not the example of the guidelines.

Response 5: Thanks again for your careful reading. Paragraphs and sections in part 3 are change to plain. And the paragrah 3.1 has been modified.

Comment 6: In Figure 2 G the DAPI results of ECM-SF scaffold and the results of light microscopy seem to have different orientation. The lines of the light microscopy image do not follow the lines of the DAPI-blue fluorescence. In all Figures more detailed description need to be provided for the visualized results. For example, in line 258-259 the authors refer to Figure 2G results by mentioning scaffolds containing dECM, however this is not provided clearly either in the Figure or within the description. The authors state at line 215 that they use DAPI for evaluating DNA existence and at line 258 that there is no blue-fluorescence, however in Figure 2G there is blue fluorescence. Figures 2G, H, and I are too small to be clearly studied and evaluated.

Response 6: Thanks for the comment. The figures of DAPI and light microscopy have been replaced. In figure 3G, the two figures in the left side are ECM-SF scaffolds, and the two figures in the right side are multi-channeled ECM-SF scaffolds. It may be easier to identified after providing the figures at their first mentioned in the main text. The figure legend has been revised to “DAPI and light microscopy images”.

There is obscure description in our manuscript. It has been revised to “no positive nuclear blue-fluorescent was detected”. The blue-fluorescent in figure 3G is the nonspecific fluorescent of scaffolds.

Comment 7: The authors should connect their results with the discussion and mention the specific figures when appropriate. For example, lines 376-377 the authors say that “As shown in the results, the double cross-linking method did not destroy the 376 dECM components in the hydrogel”. How is this supported by their results? More explanations need to be provided and proper connection of results and discussion.

Response 7: Thank you for your valuable advice. The discussion has been revised. The FTIR (Figure 3H) results showed that the positions of main peaks of SF and dECM had no significant change after secondary crosslinking, which demonstrated that the double cross-linking method did not destroy the dECM components in the hydrogel.

As showed in result of the water absorption capacity and surface wettability assay, The multi-channeled structure significantly improves the water absorption capacity of the hydrogel which could facilitate the cell seeding process.

Round 2

Reviewer 2 Report

Reviewer Comments to Author(s)

Recommendation: Accepted

 After a detailed evaluation of the manuscript after revision all the issues and questions have been addressed by the author(s) and the manuscript can be published for publication.
